# Simultaneous Comparison of Subxiphoid and Intercostal Wound Pain in the Same Patients Following Thoracoscopic Surgery

**DOI:** 10.3390/jcm11082254

**Published:** 2022-04-18

**Authors:** Yu-Wei Liu, Shah-Hwa Chou, Andre Chou, Chieh-Ni Kao

**Affiliations:** 1Division of Thoracic Surgery, Department of Surgery, Kaohsiung Medical University Hospital, Kaohsiung Medical University, Kaohsiung 807, Taiwan; shhwch@cc.kmu.edu.tw (S.-H.C.); jennykao0320@gmail.com (C.-N.K.); 2School of Medicine, College of Medicine, Kaohsiung Medical University, Kaohsiung 807, Taiwan; 3PhD Program in Environmental and Occupational Medicine, College of Medicine, Kaohsiung Medical University, National Health Research Institutes, Kaohsiung 807, Taiwan; 4Faculty of Medicine, Poznań University of Medical Sciences, 61-701 Poznań, Poland; drechou@gmail.com

**Keywords:** subxiphoid, intercostal, video-assisted thoracoscopic surgery (VATS), postoperative pain

## Abstract

There is a lack of data comparing postoperative pain after subxiphoid and intercostal video-assisted thoracoscopic surgery (VATS). Pain is an individual’s subjective experience and, therefore, difficult to compare between different individuals subjected to either procedure. This study assessed reported pain at six postoperative time points in the same patients receiving both subxiphoid and intercostal incisions for thoracic disease. Data from 44 patients who received simultaneous combined intercostal and subxiphoid VATS were retrospectively analyzed from August 2019 to July 2021. All patients received the same length of subxiphoid and intercostal incisions with or without drain placements. A numerical pain rating scale was administered on postoperative days (POD)-1, POD-2, POD-Discharge, POD-30, POD-90, and POD-180. Bilateral uniportal VATS was performed in 11 patients, and unilateral multiportal VATS was performed in 33 patients. In the unilateral VATS group, there were no differences in pain reported for both incisions in the early postoperative period. However, in the bilateral VATS group, subxiphoid wounds resulted in significantly higher pain scores on POD-1, POD-2, and POD-Discharge (*p* = 0.0003, 0.001, and 0.03, respectively). Higher late (3 and 6 months) postoperative pain was associated with intercostal incisions in both groups, as previously reported, whereas higher early (day 1, 2, and discharge) postoperative pain was more associated with subxiphoid incisions than intercostal incisions in the bilateral VATS group.

## 1. Introduction

Recent data comparing open thoracotomy to video-assisted thoracoscopic surgery (VATS) suggests a predilection for the latter. Randomized and non-randomized studies have associated VATS procedures with less trauma, decreased postoperative pain, fewer postoperative complications and faster recovery. Findings such as these have led to wider usage of VATS among thoracic surgeons [1,2,3]. Continued advances in minimally invasive thoracic surgery have permitted the evolution from 3-or 4-port VATS techniques to the uniportal technique. Uniportal VATS is becoming more popular globally due to the smaller number of incisions required and decreased postoperative chest wall neuralgia, and, as suggested by one study, it is more ergonomic than other multiport techniques [4].

It has been suggested that uniportal VATS is less painful than multiportal VATS based on the assumption that one incision is less painful than two or more. However, questions have arisen regarding the relative degree of postoperative pain and chest wall neuralgia among patients receiving uniportal VATS compared with those who receive multiportal VATS [5,6,7,8]. While VATS is a minimal access surgical technique, as much as one-third of patients receiving these procedures report chronic pain (pain lasting 2 to 3 months) possibly due to intercostal nerve compression [9]. It is reasonable to assume then that subxiphoid uniportal VATS, an approach which avoids the intercostal spaces, would reduce postoperative pain.

In 2012, Suda et al. reported their first concomitant use of subxiphoid single-port thymectomy and CO_2_ insufflation [10]. In 2014, Liu et al. reported their first use of subxiphoid uniportal VATS lobectomy [11]. Subsequently, this alternative approach started being used to treat a wide array of thoracic procedures worldwide [12,13,14,15,16]. This relatively new technique often requires division of the rectus abdominis muscle, resection of the xiphoid process and use of a sternal elevator. However, these additional manipulations might potentially offset any pain advantages gained from not injuring the intercostal nerves [17]. Moreover, proponents of its use have admitted that there may be some cardiac compression when using subxiphoid VATS to perform left thoracic procedures as well as limited visualization of the posterior mediastinum and potential technical difficulties [11,12,13,14,15,16].

While theoretically there should be less pain, is there actually less pain? Pain is a subjective and multidimensional experience. One of the most challenging aspects of any pain research is the objective evaluation of pain. Although several studies have been performed to investigate the incidence and intensity of pain after thoracic surgery [9,18,19,20,21,22,23], to the best of our knowledge, none have used the same patients to investigate postoperative pain associated with both subxiphoid and intercostal incisions. A comparison of the postoperative pain associated with the two approaches in the same patients should reduce some bias arising from comparing use of one approach on one group of patients with the use of the other approach on a different group of patients.

Therefore, this study accessed reported pain associated with both subxiphoid and intercostal incisions of the same lengths on the same thoracic disease patients. Pain was rated numerically at six different postoperative time points, three early and three late.

## 2. Methods

### 2.1. Study Design

This retrospective cohort study contained prospectively collected data. It was approved by the research ethics committee at Kaohsiung Medical University Hospital (Approval number KMUHIRB-E(I)-20200228). The requirement for written informed consent was waived. Fifty-six patients receiving combined subxiphoid and intercostal VATS for thoracic procedures at a single medical center from August 2019 to July 2021 were consecutively enrolled. All incisions were equal in length. None of the patients had preoperative analgesic requirements, previous thoracic surgeries, impaired cardiac functioning, or body mass indices (BMI) >30 kg/m^2^. Five patients whose subxiphoid and intercostal incisions were not the same length, three whose postoperative drains were not the same size, and four who received intraoperative thoracic epidural anesthesia were excluded. After exclusion, the remaining 44 patients were divided into two groups, (1) eleven patients receiving simultaneous bilateral uniportal VATS (*n* = 11) with ipsilateral subxiphoid incisions on one side and contralateral intercostal incisions on the other and (2) thirty-three patients receiving unilateral multiportal VATS (*n* = 33) with both surgical approaches used on the same side (Figure 1).

All procedures were performed by one surgical team following the same perioperative protocols. Patient data, including age, gender, BMI, smoking habits, lung function tests, perioperative data, postoperative complications, and follow-up pain scores were collected from digital medical records. 

### 2.2. Operative Technique

#### 2.2.1. Simultaneous Bilateral Uniportal VATS (11 Patients)

Under general anesthesia via a double-lumen endotracheal tube, nine of the simultaneous bilateral uniportal VATS patients received right side subxiphoid VATS pulmonary wedge resections and left side intercostal VATS pulmonary wedge resections. The other two patients received right side intercostal VATS upper lobectomies and left side subxiphoid VATS lower lobe wedge resections (See Figure 2 and Table 1).

Surgical equipment included Echelon Flex endoscopic articulating linear cutters (Ethicon Endo-Surgery, Somerville, NJ, USA), long curved dissector and grasping forceps with double articulations, and curved suction (Scalan International, Inc. Saint Paul, MN, USA), and a 10 mm 30° thoracoscope (Karl Storz, Munich, Germany). Patients were placed in a supine position on a tilted operating table (30 to 45 degrees) for easy access and performance of subxiphoid VATS surgery. A single incision 2–5 cm long was made below the xiphoid process and slightly slanted along the costal arch. The skin, subcutaneous tissue, and rectus abdominis were cut, blunt dissection was performed using a finger applied to the deep surface of the costal arch, and the pleural cavity was accessed. A plastic wound protector (Alexis, Applied Medical, CA, USA) was placed in the incision. Throughout the procedure, the xiphoid process was not excised and no sternal elevator was used. After uniportal subxiphoid VATS pulmonary wedge resection, a chest drain was inserted through the same incision. For the other side, the patient was changed to a decubitus position to access the contralateral lung. A single incision with the same length as the subxiphoid incision was made over the intercostal space, and a wound protector was inserted. A chest drain the same size as used on the other side was placed through the intercostal incision (Figure 3A).

#### 2.2.2. Unilateral Multiportal VATS (33 Patients)

Following similar anesthetic procedures and positioning for the optimized subxiphoid approach, multiportal combined subxiphoid and intercostal VATS were performed depending on surgical procedure type (See Figure 4 and Table 2).

Incisions equal in length were made over the subxiphoid and intercostal regions. For patients with pulmonary or mediastinal tumors, specimens were retrieved through the subxiphoid incision without enlarging the wound. A total of 15 of the 33 patients received an additional small third port (0.5–1 cm) for intraoperative manipulation and placement of postoperative chest drains (Figure 3B). The other 18 patients received 2-port VATS, with some patients (*n* = 3) receiving two equal but separate drains via the subxiphoid and intercostal incisions. In total, 15 patients received no chest drain.

### 2.3. Postoperative Management

Chest radiography was performed immediately after the operation or the following morning. Chest drains were removed in both groups if there were no air leaks and if the drainage was < 200 mL within 24 h. A total of 15 of the 44 patients received intravenous patient-controlled anesthesia upon request. Acetaminophen, diclofenac, and tramadol were regularly administered once patients resumed normal oral intake until discharge. Additional doses of intravenous parecoxib were used for intolerable pain during the postoperative hospital stay. Pain scores for both subxiphoid and intercostal wounds were simultaneously assessed separately by a surgical team member using a numerical rating scale (NRS), 0 (no pain) to 10 (excruciating pain), every eight hours with patient at rest on postoperative day 1 (POD-1) and then on subsequent postoperative days until discharge. The mean of the three daily NRS scores for both incisions were recorded for the three early time points. Pain scores at the three late time points were followed up either by a consultation in the outpatient clinic or a phone call.

### 2.4. Statistical Analysis

Descriptive data were expressed as numbers with percentages and continuous variables expressed as either means with standard deviation or medians with interquartile range (IQR). The Student’s *t*-test (paired *t*-test) was used to compare subxiphoid and intercostal NRS scores from time point to time point among the same patients. All statistical operations were performed using SPSS 20 (SPSS Inc., Chicago, IL, USA) for Windows. All statistical tests were two-tailed, and a *p*-value <0.05 was considered significant.

## 3. Results

Of the 44 patients receiving subxiphoid VATS for the surgical treatment of thoracic conditions between August 2019 and July 2021, 11 simultaneously received uniportal ipsilateral subxiphoid VATS on one side and contralateral intercostal VATS on the other side, while 33 received multiportal subxiphoid and intercostal VATS on one side only. As can be seen in Table 3, males made up fifty-nine percent of the patients, and the mean BMI was 23 kg/m^2^. Other perioperative variables such as operative time, blood loss, complications, and postoperative hospital length-of-stay were comparable and similar to our own experience with transthoracic VATS.

As can be seen in Table 1, out of the eleven patients receiving bilateral VATS, five had primary spontaneous pneumothorax with contralateral blebs for which they received surgical treatments following previously reported procedures [24], and six had indeterminate lung nodules or suspected lung cancer and received simultaneous bilateral operations for accurate diagnosis and staging after weighing relative risks and benefits. Of the thirty-three patients who received unilateral VATS, two lung cancer patients who had malignant pericardial effusion received pericardial windows, and the remaining thirty-one had pulmonary or mediastinal tumors (see Table 2). Based on our previous experience, postoperative drains were not placed in patients following mediastinal tumor resection [25]. For patients who had received pulmonary lobectomies and wedge resections and who were at risk of air leaks, chest drains were positioned via a small third port incision. In both groups, all subxiphoid and intercostal incisions and drains, when created, were equal in incision length and drain size (Figure 5 and Figure 6).

Table 4 shows the mean subxiphoid incision and intercostal incision NRS pain score results for three postoperative hospital days, postop days 1, 2, and discharge, and three postoperative follow-up days: 30, 90, and 180. In patients receiving bilateral VATS (on two sides), subxiphoid incisions were associated with significantly higher mean pain scores than the intercostal incisions in the early postoperative period (POD-1, POD-2, and POD-Discharge (*p* = 0.0003, 0.001, and 0.03, respectively)) but lower mean pain scores on POD-90 and POD-180 (*p* = 0.03 and 0.16) (Figure 7A). Interestingly, in patients receiving unilateral VATS, the pain score differences between the two incisions were insignificant in the early postoperative period. Only intercostal incisions appeared significantly higher on POD-90 and POD-180 (*p* = 0.03 and 0.08) (Figure 7B). Comparisons between the mean subxiphoid incision and intercostal incision NRS pain scores for overall patients are shown in Appendix A.

## 4. Discussion

Pain is usually assessed subjectively by means of verbal or visual intensity scales and questionnaires [9,21]. Most pain scales are subjective assessment tools because pain is interpreted differently by each individual. In this study, diligent attempts to control as many confounding variables as possible were made. These attempts included subjective individuality. This was achieved by performing two incisions on the same patients during the same operation session and asking them to rate their incision-related pain at the same six postoperative time points. Incision length, chest tube size and perioperative analgesic protocols were all controlled for. Careful attention toward reducing the influence of these potential confounders should reduce their effect on results and produce a more reliable comparison of pain following thoracic surgery.

In contrast to the literature’s overwhelming opinion that subxiphoid VATS causes much less early postoperative pain than intercostal VATS following thoracic surgery [12,13,14,15,16,18], our study found the subxiphoid incision produced more pain than the intercostal incision during the early postoperative period (POD-1, POD-2, and POD-Discharge), but only in the bilateral VATS group, not in the unilateral VATS group. One reason for the difference may be that the subxiphoid incision was always made below the xiphoid process and slightly tilted in the direction of the operative costal arch. There was a greater distance between the two incisions (subxiphoid and intercostal) in the bilateral VATS group than in the unilateral VATS group, which could possibly decentralize a patient’s focus of attention. Another reason may be that 15 of the 33 patients in the unilateral VATS group received 3-port VATS. The chest drains were placed through a small intercostal port, which has been reported to have a negative impact on pain [20]. This would also confound perceptions of pain for other incisions. These two factors could interfere with a clear comparison of pain between subxiphoid and intercostal incisions in the unilateral VATS group.

Although the findings of this study were in contrast to findings in the literature (less early postoperative pain in subxiphoid compared to intercostal VATS), the present study did find higher pain intensity associated with intercostal incisions in the late follow-up period (POD-90 and 180), consistent with the literature [18]. None of the eleven patients receiving bilateral VATS reported chronic pain (pain lasting 2 to 3 months) over the subxiphoid incision, while four (4/11) reported pain over the intercostal incision on POD-90, and two reported pain through until POD-180. Furthermore, 1 of the 33 patients receiving unilateral VATS reported pain over both subxiphoid and intercostal incisions and 7 reported pain over the intercostal incisions on POD-90. The three patients reporting on POD-90 continued to report the same pain on POD-180. It is interesting to note that 9 of the 12 patients who reported chronic pain in our cohort study had primary lung cancers, lung metastases, or thymic malignancies for which they received adjuvant chemotherapy and/or radiotherapy within six months of surgery. Two of the twelve had pulmonary tuberculosis, for which they received six months of anti-tuberculosis therapy, and the other received an unexpected surgical intervention for acute cerebrovascular disorder within three months following thoracic surgery. Previous studies have reported that pain can be expressed or experienced differently depending on race, gender, age, and treatment modalities, including radiotherapy, pleurectomy, more extensive surgical procedures, and other multifactorial mechanisms [19,23]. Recently, Yoon and colleagues also reported that adjuvant chemotherapy serves as a risk factor for chronic pain in lung cancer patients after VATS [26], supporting the findings of this study.

The use of subxiphoid VATS incisions avoids intercostal nerve damage, which has often been cited as a major source of postoperative pain and paresthesia after thoracotomy and intercostal VATS. From a surgical standpoint, this approach allows easier access to the anterior mediastinum when performing thymectomy as well as better access to both sides of the chest with only a single incision. However, this approach, particularly when there is posterior mediastinum and left thorax involvement, can be challenging even for experienced thoracic surgeons. Whether the potential advantages outweigh the potential disadvantages remains to be determined by future studies.

Reviewing the literature, we found it took a long time for the field to draw a firm conclusion regarding relative reported pain associated with thoracotomy versus VATS [1,2,3,9,19]. The debate regarding the relative pain associated with multiportal VATS versus uniportal VATS is also ongoing [5,6,7,8]. Currently, only a few studies have reported a reduction in acute postoperative pain and chronic pain associated with subxiphoid VATS [12,13,14,15,16,18].

This study has a number of limitations. Our sample size was small. The difficulty in recruiting patients who required receiving two procedures in the same locations at the same time made this limitation unavoidable. Another limitation is that all the procedures in this study were performed by one surgeon at one institute, and so the learning curve for the performance of subxiphoid VATS has not been taken into consideration. In addition, when considering our results, the non-constant confounding posed by the use of different types of anaesthesia and analgesics with variable doses in different groups should be considered. Therefore, further studies, including well-controlled prospective studies and multi-center studies, are needed to further verify our findings.

## 5. Conclusions

Our study found that patients who received simultaneous uniportal ipsilateral subxiphoid VATS and contralateral intercostal VATS reported more pain associated with subxiphoid incisions than intercostal incisions. Thus, the use of subxiphoid VATS may not always result in greater pain reduction than intercostal incisions in the early postoperative period. Additionally, the higher pain intensity in the intercostal incisions reported in the late postoperative period may not be indicative of pain caused by merely nerve damage during surgery, as there were other potential multifactorial pain-causing or pain-worsening mechanisms such as adjunctive chemotherapy and/or radiotherapy involved. More research is needed to better understand the cause of pain in a surgical area previously not thought to be prone to pain.

## Figures and Tables

**Figure 1 jcm-11-02254-f001:**
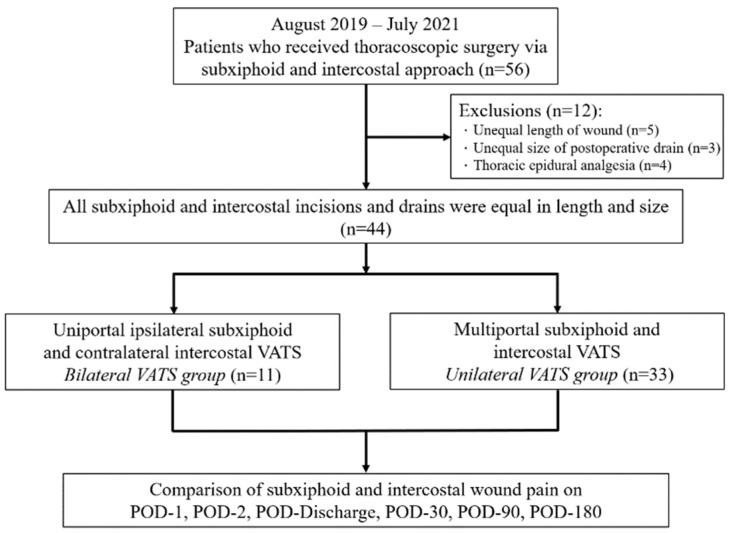
Flow diagram for patient recruitment. VATS, video-assisted thoracoscopic surgery; POD, postoperative day.

**Figure 2 jcm-11-02254-f002:**
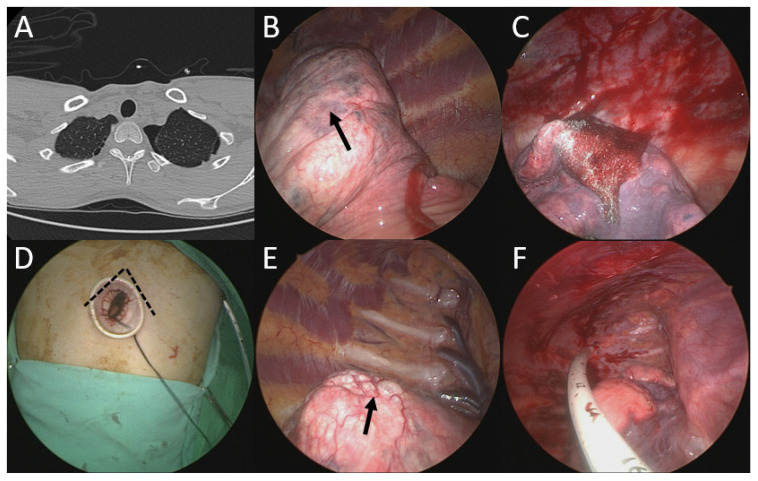
A patient with left PSP and right contralateral blebs underwent simultaneous uniportal left intercostal and right subxiphoid VATS wedge resection and pleurodesis. (**A**) Computed tomography scan showed left PSP and right contralateral blebs. (**B**) Emphysema-like changes observed over left upper lobe (arrowhead). (**C**) After left upper lobe wedge resection and staple lines covered with reinforcement felt. (**D**) Uniportal subxiphoid VATS for right side procedure. (**E**) Right upper lobe blebs observed (arrowhead). (**F**) Chest drain the same size as used on the other side was placed through the subxiphoid incision. PSP, primary spontaneous pneumothorax. VATS, video-assisted thoracoscopic surgery.

**Figure 3 jcm-11-02254-f003:**
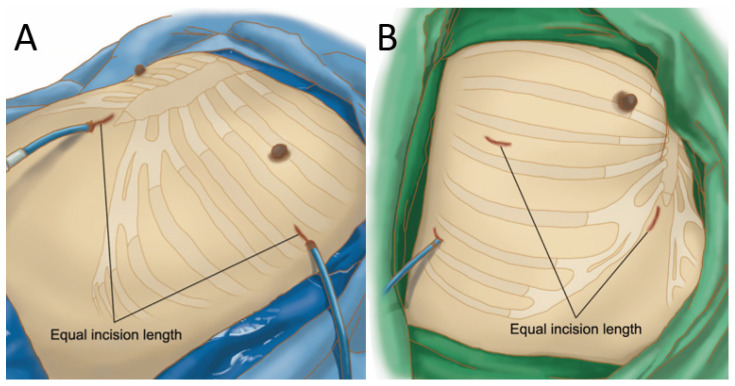
Postoperative view of subxiphoid and intercostal incision. All subxiphoid and intercostal incisions and drains were equal in length and size, regardless of whether they were used in bilateral uniportal VATS (**A**) or unilateral multiportal VATS (**B**).

**Figure 4 jcm-11-02254-f004:**
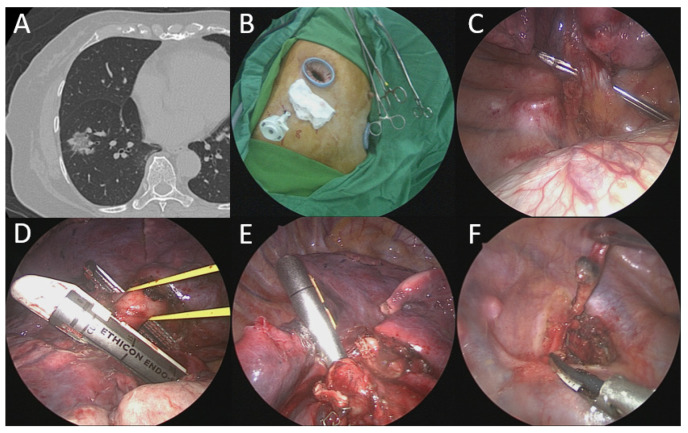
A patient with right lower lung cancer receiving unilateral 3-port VATS right lower lobectomy. (**A**) Computed tomography scan showed a right lower lung part-solid ground-glass nodule (2.0 cm). (**B**) Equal-length incisions (3 cm) made over subxiphoid and 5th intercostal space as well as another 1-cm incision over 7th intercostal space. (**C**) Inferior pulmonary vein identified and transected via subxiphoid incision. (**D**) Basal trunk of pulmonary artery transected by endostapler via subxiphoid incision. (**E**) Right lower lobe bronchus transected by endostapler via subxiphoid incision. (**F**) Upper mediastinal lymph node dissection using harmonic scalpel from 5th intercostal incision.

**Figure 5 jcm-11-02254-f005:**
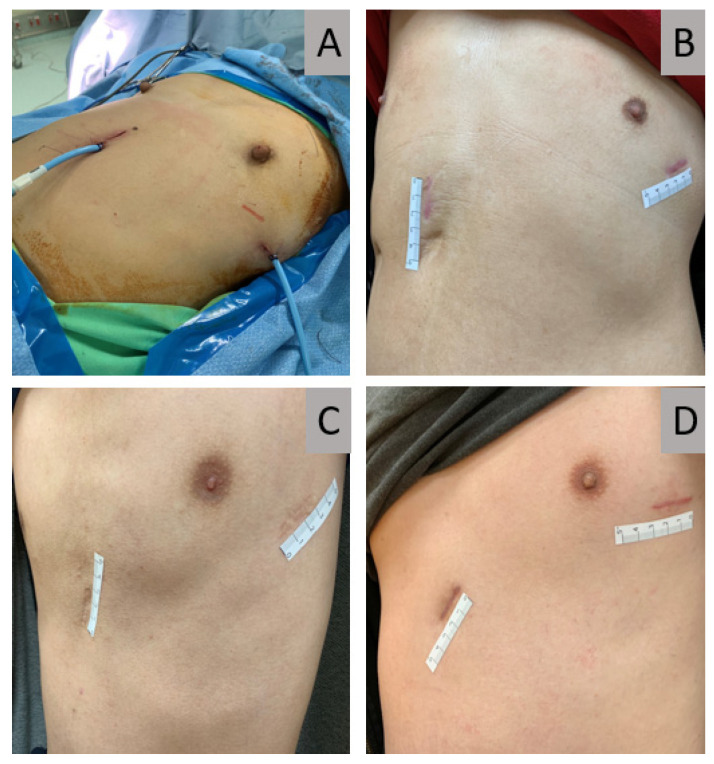
Patients receiving simultaneous uniportal ipsilateral subxiphoid and contralateral intercostal VATS. (**A**) A patient with equal length of subxiphoid and intercostal incision and drain size. (**B**–**D**) Postoperative pictures of three different patients with equally long subxiphoid and intercostal incisions three months following surgery. VATS, video-assisted thoracoscopic surgery.

**Figure 6 jcm-11-02254-f006:**
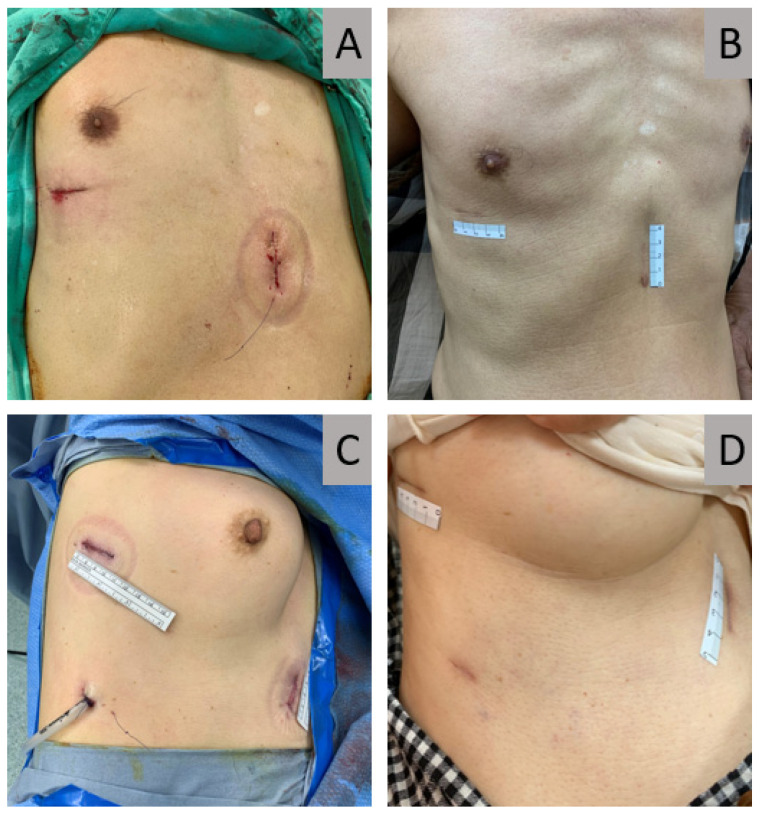
Patients receiving multiportal subxiphoid and intercostal VATS. (**A**) A patient with anterior mediastinal tumor received two-port VATS resection with subxiphoid and intercostal incision with equal lengths and no postoperative drain insertion. (**B**) Postoperative picture of the above-mentioned patient 3 months following surgery. (**C**) A patient with lung cancer received 3-port VATS right lower lobectomy with subxiphoid and intercostal incisions of the same lengths and drain-insertion via the small 7th intercostal wound. (**D**) Postoperative picture of the above-mentioned patient 3 months following surgery. VATS, video-assisted thoracoscopic surgery.

**Figure 7 jcm-11-02254-f007:**
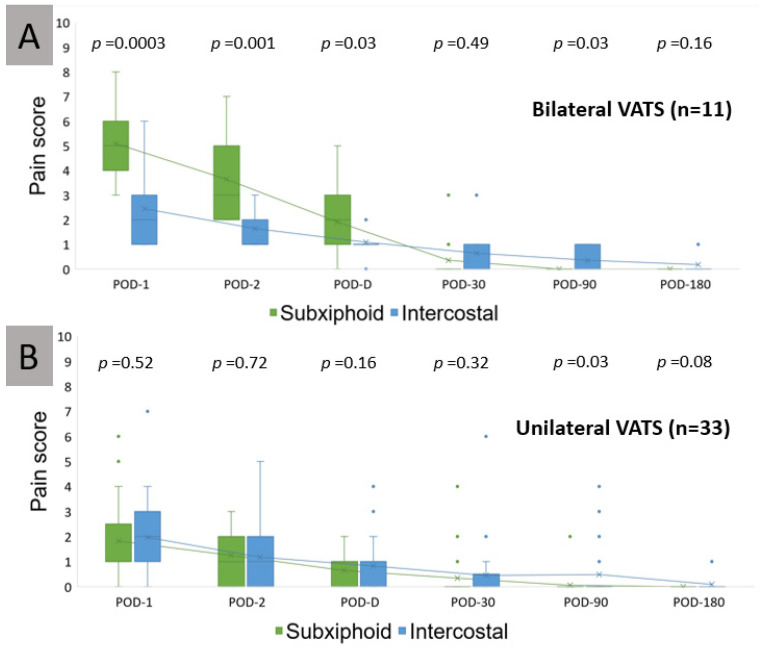
Box and whisker plots showing pain scores for the subgroups of patients. Box plots showing pain scores for subxiphoid and intercostal incisions in the same patient receiving bilateral uniportal VATS (**A**) and unilateral multiportal VATS (**B**). Arrowhead, equal length subxiphoid and intercostal incisions. VATS, video-assisted thoracoscopic surgery. Subxiphoid incision represented in green, intercostal in blue. POD, postoperative day. D, discharge.

**Table 1 jcm-11-02254-t001:** Perioperative details of patients with bilateral VATS (*n* = 11).

Case No.	Age	Sex	Procedure Type (Sub vs. ICS)	Port	Drain Size (Sub vs. ICS) (Fr)	Incision Size (Sub vs. ICS) (cm)	Operative Time (Sub vs. ICS) (min)	Blood Loss (Sub vs. ICS) (ml)	Pathology (Sub vs. ICS)
1	58	F	Wedge resection (RUL vs. LLL)	1-port	16/16	3.0/3.0	50/40	5/5	PLC
2	23	M	Wedge resection (RUL vs. LUL)	1-port	24/24	2.5/2.5	90/70	20/10	Bullae
3	56	M	Wedge resection (RML vs. LLL)	1-port	16/16	3.0/3.0	50/50	5/5	Metastatic RCC
4	55	M	Wedge resection (LLL) vs. lobectomy (RUL)	1-port	24/24	4.0/4.0	70/150	20/80	PLC/Subpleural LN
5	18	M	Wedge resection (RUL vs. LUL)	1-port	24/24	2.5/2.5	40/40	20/20	Bullae
6	25	M	Wedge resection (RUL vs. LUL)	1-port	12/12	2.5/2.5	40/30	5/5	Bullae
7	67	M	Wedge resection (LLL) vs. lobectomy (RUL)	1-port	14/14	4.0/4.0	50/150	5/15	PLC/AIS
8	28	M	Wedge resection (RUL vs. LUL)	1-port	12/12	2.5/2.5	50/50	5/5	Bullae
9	54	F	Wedge resection (RLL vs. LLL)	1-port	14/14	3.0/3.0	60/90	5/5	Sarcoidosis
10	57	F	Wedge resection (RLL vs. LLL)	1-port	12/12	3.0/3.0	70/70	5/5	Tuberculosis
11	18	M	Wedge resection (RUL vs. LUL)	1-port	12/12	2.0/2.0	70/60	10/15	Bullae

VATS, video-assisted thoracoscopic surgery; Sub vs. ICS, subxiphoid vs. intercostal approach; RUL, right upper lobe; RML, right middle lobe; RLL, right lower lobe; LUL, left upper lobe; LLL, left lower lobe; PLC, primary lung cancer; RCC, renal cell carcinoma; LN, lymph node; AIS, adenocarcinoma in situ.

**Table 2 jcm-11-02254-t002:** Perioperative details of patients with unilateral VATS (*n* = 33).

Case No.	Age	Sex	Procedure Type	Port	Drain Size (Sub vs. ICS) (Fr)	Incision Size (Sub vs. ICS vs. 3rd Port) (cm)	Operative Time (min)	Blood Loss (mL)	Pathology
1	64	M	Mediastinal tumor resection	2-port	No drain	3.0/3.0	90	20	Thymic hyperplasia
2	76	F	Mediastinal tumor resection	2-port	No drain	3.0/3.0	60	5	Thymoma
3	46	F	Wedge resection (RML)	2-port	No drain	2.5/2.5	60	5	Metastasizing leiomyoma
4	48	F	Wedge resection (RUL, RLL)	2-port	15/15	3.0/3.0	100	20	Tuberculosis
5	71	F	RLL lobectomy	3-port	14 (*)	4.0/4.0/1.0	150	15	Primary lung cancer
6	60	F	RUL lobectomy	3-port	20 (*)	3.0/3.0/1.0	170	30	Primary lung cancer
7	58	M	Wedge resection (LUL)	3-port	24 (*)	3.0/3.0/1.0	120	10	Metastasis of nasopharyngeal cancer
8	57	M	Pericardial window	3-port	12 (*)	2.5/2.5/0.5	90	5	Metastasis of primary lung cancer
9	57	M	RUL lobectomy	3-port	28 (*)	5.0/5.0/1.0	210	200	Primary lung cancer
10	56	M	Pericardial window	3-port	12 (*)	2.5/2.5/0.5	40	5	Metastasis of primary lung cancer
11	64	M	RLL lobectomy	3-port	24 (*)	4.0/4.0/1.0	180	60	Primary lung cancer
12	65	F	Wedge resection (RUL, RLL)	3-port	12 (*)	2.5/2.5/0.5	60	5	Metastasis of thymic carcinoma
13	52	M	Wedge resection (RLL)	2-port	No drain	2.5/2.5	60	5	Metastasis of renal cell carcinoma
14	61	F	Wedge resection (RLL)	3-port	15 (*)	2.5/2.5/0.5	60	5	Organizing pneumonia
15	54	M	RUL lobectomy	2-port	14/14	3.0/3.0	170	50	Primary lung cancer
16	63	F	Mediastinal tumor resection	3-port	16 (*)	2.5/2.5/0.5	130	10	Thymoma
17	56	F	Mediastinal tumor resection	3-port	No drain	2.5/2.5/1.0	110	5	Thymic cyst
18	63	M	Mediastinal tumor resection	3-port	No drain	3.0/3.0/1.0	120	20	Thymolipoma
19	58	F	Mediastinal tumor resection	3-port	16 (*)	2.5/2.5/0.5	130	10	Thymoma
20	18	M	Mediastinal tumor resection	3-port	No drain	2.5/2.5/1.0	100	10	Thymic hyperplasia
21	45	F	Mediastinal tumor resection	3-port	No drain	3.0/3.0/1.0	120	5	Thymoma
22	55	F	Mediastinal tumor resection	3-port	No drain	3.0/3.0/1.0	130	5	Thymoma
23	44	M	Mediastinal tumor resection	3-port	No drain	3.0/3.0/1.0	120	10	Thymoma
24	54	F	Mediastinal tumor resection	3-port	No drain	3.0/3.0/1.0	100	5	Thymic cyst
25	48	M	Mediastinal tumor resection	3-port	16 (*)	3.0/3.0/1.0	150	30	Atypical carcinoid
26	52	M	Mediastinal tumor resection	3-port	No drain	2.5/2.5/0.5	110	10	Thymoma
27	51	F	Mediastinal tumor resection	3-port	No drain	2.5/2.5/0.5	90	5	Thymoma
28	43	M	Mediastinal tumor resection	2-port	No drain	3.0/3.0	110	5	Thymoma
29	73	M	Mediastinal tumor resection	3-port	16 (*)	3.0/3.0/0.5	100	10	Angiolipoma
30	47	F	Mediastinal tumor resection	3-port	16 (*)	2.5/2.5/0.5	180	30	Thymic carcinoma
31	62	M	Mediastinal tumor resection	2-port	No drain	3.0/3.0	100	5	Thymic hyperplasia
32	39	M	Mediastinal tumor resection	3-port	16 (*)	2.5/2.5/0.5	150	10	Thymoma
33	54	M	Mediastinal tumor resection	2-port	14/14	3.0/3.0	170	10	Thymoma

VATS, video-assisted thoracoscopic surgery; Sub vs. ICS, subxiphoid vs. intercostal approach; RUL, right upper lobe; RML, right middle lobe; RLL, right lower lobe; LUL, left upper lobe; (*) stands for chest-drain insertion through the third port.

**Table 3 jcm-11-02254-t003:** Characteristics of enrolled patients (*n* = 44).

Characteristic	Value
Mean age (range), y	52 (18–76)
Gender, % (*n*)	
Male	59 (26)
Female	41 (18)
Smoking (yes), % (n)	34 (15)
Mean BMI (range), kg/m^2^	23 (16.5–30)
Pulmonary function test	
Mean FEV1 (range), L	2.5 (1.6–3.7)
Mean FEV1 (range), Predicted %	86 (64–117)
Grade I-II complication, % (*n*)	11.3 (5)
Prolonged air leak (>5 days)	4.5 (2)
Atrial fibrillation	2.3 (1)
Wound allergy	2.3 (1)
Wound poor healing	2.3 (1)
Mean postoperative stay (range), day	4 (2–9)
Median wound length (range), cm	3.0 (2.0–5.0)
Median drain size (range), Fr	14 (12–28)
Median operation time (range), min	90 (40–240)
Median blood loss (range), ml	10 (5–200)

BMI, body mass index; FEV1, forced expiratory volume in the first second of expiration.

**Table 4 jcm-11-02254-t004:** Postoperative numerical rating scale (NRS) pain scores for patient subgroups.

Pain Score	Bilateral VATS (*n* = 11)					Unilateral VATS (*n* = 33)				
	Subxiphoid Wound	95% CI	Intercostal Wound	95% CI	*p*-Value	Subxiphoid Wound	95% CI	Intercostal Wound	95% CI	*p*-Value
POD-1	5.1 ± 1.4	(4.1–6.0)	2.5 ± 1.4	(1.5–3.4)	0.0003	1.8 ± 1.5	(1.3–2.4)	2.0 ± 1.6	(1.4–2.5)	0.52
POD-2	3.6 ± 1.7	(2.5–4.8)	1.6 ± 0.8	(1.1–2.2)	0.001	1.2 ± 1.1	(0.8–1.6)	1.2 ± 1.4	(0.7–1.7)	0.72
POD-Discharge	1.9 ± 1.4	(1.0–2.8)	1.1 ± 0.5	(0.7–1.5)	0.03	0.6 ± 0.7	(0.4–0.9)	0.8 ± 1.0	(0.5–1.2)	0.16
POD-30	0.4 ± 0.9	(0.0–1.0)	0.6 ± 0.9	(0.0–1.3)	0.49	0.3 ± 0.8	(0.0–0.6)	0.5 ± 1.1	(0.0–0.9)	0.32
POD-90	0 ± 0	(0.0–0.0)	0.4 ± 0.5	(0.0–0.7)	0.03	0.1 ± 0.3	(0.0–0.2)	0.5 ± 1.1	(0.1–0.9)	0.03
POD-180	0 ± 0	(0.0–0.0)	0.2 ± 0.4	(0.0–0.5)	0.16	0 ± 0	(0.0–0.0)	0.1 ± 0.3	(0.0–0.2)	0.08

NRS, numerical rating scale; VATS, video-assisted thoracoscopic surgery; CI, confidence interval; POD, postoperative day. Data were expressed as mean ± standard deviation.

## Data Availability

Not applicable.

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
