# Peer review of "Simultaneous Comparison of Subxiphoid and Intercostal Wound Pain in the Same Patients Following Thoracoscopic Surgery"

_jcm, 2022, doi:10.3390/jcm11082254_

Round 1

Reviewer 1 Report

The changes made to the manuscript give more quality to the communication and eliminate possible misunderstandings for the reader.
The tables and figures are clear and accompanied by comprehensive notes.
Only a few typos and style/minor spell checks are required.
Congratulations to the authors for the excellent work.

Author Response

Thank you for your comment. We have gone through the manuscript carefully and have had it edited by a native English speaker with experience editing medical papers again.

Reviewer 2 Report

It has been revised as I requested.

Author Response

Thank you for your comment.

Reviewer 3 Report

The study is interesting compares pain intensity in patients having video assisted thoracic procedures with a comparison between subxiphoid and intercostal incisions in the same patient. Several minor issues need to be addressed:

  1. In abstract: "as have been reported by previous,", this should not be in abstract.
  2. In abstract: "whereas have higher early (Day 1, 2, and Discharge) postoperative pain was found subxiphoid incisions " The statement needs English editing.
  3. What is the type of the study? Enrollment of patients ended in July 2021 with 6 months follow up ending in January or February 2022, How could it be a retrospective one as mentioned for waiving the informed consent.
  4. Different types of anaesthesias and of analgesics with variable doses constitute non-constant confounders to the comparison among groups. This should be mentioned in the study limitations. 
  5. In results: "Operative time, blood loss, complications, and postoperative hospital length-of-stay were comparable and similar to those reported in the subxiphoid VATS literature [12–16]." The comparison with the literature should be in discussion not results.

Thank you

Author Response

Thank you for your comments.

(1) We have deleted reference to previous studies in the abstract.

(2) We have also edited the English in the sentence on postoperative pain, which now reads, “Higher late (3 and 6 months) postoperative pain was associated with intercostal incisions in both groups, whereas higher early (Day 1, 2, and Discharge) postoperative pain was more associated with subxiphoid incisions than intercostal incisions in the bilateral VATS group.”

(3) We have attempted to explain the kind of study. That sentence now reads as follows: This retrospective cohort study containing prospectively collected data. It was approved by the research ethics committee at Kaohsiung Medical University Hospital (Approval number KMUHIRB-E(I)-20200228). Requirement for written informed consent was waived.

(4) We have added the confounding due to different anaesthesias and of analgesics to the limitations in a sentence that reads, “In addition, when considering our results, the non-constant confounding posed by the use of different types of anaesthesia and analgesics with variable doses in different groups should be considered.”

(5) Regarding the sentences explaining why do not present unremarkable findings in the text of the results, we added a sentence that says, “Other perioperative variables such as operative time, blood loss, complications, and postoperative hospital length-of-stay were expected and similar to our own experience with transthoracic VATS.” We have moved reference to similar findings in other studies to the paragraph of discussion. That sentence read, “Operative time, blood loss, complications, and postoperative hospital length-of-stay were comparable and similar to those reported in the subxiphoid VATS literature [12–16].

Again, thank you. Your suggestions have really helped us improve the presentation of information in our manuscript.

This manuscript is a resubmission of an earlier submission. The following is a list of the peer review reports and author responses from that submission.

Round 1

Reviewer 1 Report

C and D in fig.3 are the contents that should be in the result, and they are in the method part. This confuses my understanding of the content. It is better to place the fig. and table related to the result in the result part, and the figure related to the method in the method part.

Reviewer 2 Report

This is a retrospective study to assess postoperative pain after VATS, comparing two different trocar insertion positions.
The study does not have a significant impact considering what already exists in the literature and the level of evidence presented is low because it is not a randomized clinical trial.

Reviewer 3 Report

Dear Author

This manuscript is very interesting.

However, there are big bias.

This study design is quite  ununderstandable. If patient have costal incision, this patient usually have neuralgia pain around subxiphoid. Therefore, assessing pain is quite difficult for both incision.

I think this manuscript is not suitable for publish.

Thanks,

Reviewer 4 Report

It is difficult to compare pain in patients that have more than one skin incision and it could affects the results. I found a lot of english mistakes

Reviewer 5 Report

Compliments to the Authors.

This exciting study presents preliminary data from a small cohort of patients, evaluating pain scores discrepancies between different surgical approaches for thoracic surgery procedures.In particular, the patient feedback in pain after the subxiphoid approach has been compared to the intercostal approach in the same patients.
From the mere point of statistical significance, the main limitation referred to the small series may be considered overpassed by the quality of the study design and the presentation of the results.

A minor language revision could foment more interest to the readers.